# ApoE4 Is Associated with Lower Body Mass, Particularly Fat Mass, in Older Women with Cognitive Impairment

**DOI:** 10.3390/nu14030539

**Published:** 2022-01-26

**Authors:** Takafumi Ando, Kazuaki Uchida, Taiki Sugimoto, Ai Kimura, Naoki Saji, Shumpei Niida, Takashi Sakurai

**Affiliations:** 1Center for Comprehensive Care and Research on Memory Disorders, National Center for Geriatrics and Gerontology, Obu 474-8511, Aichi, Japan; takafumi.ando@aist.go.jp (T.A.); uchida.k@ncgg.go.jp (K.U.); taiki-s@ncgg.go.jp (T.S.); sajink@ncgg.go.jp (N.S.); 2Human Behavior Research Team, Human-Centered Mobility Research Center, Information Technology and Human Factors, National Institute of Advanced Industrial Science and Technology, Tsukuba 305-8561, Ibaraki, Japan; 3Department of Public Health, Graduate School of Health Sciences, Kobe University, Kobe 654-0142, Hyogo, Japan; 4Department of Prevention and Care Science, Research Institute, National Center for Geriatrics and Gerontology, Obu 474-8511, Aichi, Japan; 5Medical Genome Center, National Center for Geriatrics and Gerontology, Obu 474-8511, Aichi, Japan; akimura@ncgg.go.jp; 6Research Institute, National Center for Geriatrics and Gerontology, Obu 474-8511, Aichi, Japan; sniida@ncgg.go.jp; 7Department of Cognitive and Behavioral Science, Graduate School of Medicine, Nagoya University, Nagoya 466-8550, Aichi, Japan

**Keywords:** ApoE, fat-free mass, fat mass, Alzheimer’s disease, cross-sectional study, outpatient study

## Abstract

A lower body mass is associated with the progression of Alzheimer’s disease (AD) and the risk of mortality in patients with AD; however, evidence of genetic determinants of decreased body mass in cognitively impaired older adults is limited. We therefore investigated the genetic effect of APOE-ε4 on body composition in older adults with mild cognitive impairment (MCI) and early-to-moderate-stage AD. A total of 1631 outpatients (aged 65–89 years) with MCI and early-to-moderate-stage AD were evaluated for the association between body composition and APOE-ε4 status. After adjusting for covariates, including cognitive function evaluated with the Mini-Mental State Examination, the presence of the APOE-ε4 was associated with lower weight (β = −1.116 ± 0.468 kg per presence, *p* = 0.017), fat mass (β = −1.196 ± 0.401 kg per presence, *p* = 0.003), and percentage of body fat (β = −1.700 ± 0.539% per presence, *p* = 0.002) in women but not in men. Additionally, the impact of APOE-ε4 on measures of body composition in women was more remarkable in MCI than in AD patients. The presence of the APOE-ε4 allele was associated with lower fat mass, particularly in women with MCI, independent of cognitive decline.

## 1. Introduction

A lower or decreasing body mass is associated with the progression of Alzheimer’s disease (AD) and the risk of mortality in patients with AD [1,2,3,4]. To date, no genetic determinants of decreased body mass have been identified in cognitively impaired older adults. Some studies have provided evidence of an association between APOE-ε4 and body mass, for example, body mass index (BMI) [5,6,7] in older adults with mild cognitive impairment (MCI) and AD. Bell et al. [5] reported that the presence of APOE-ε4 was associated with lower BMI in individuals with MCI but not in cognitively normal individuals; however, the impact of APOE-ε4 on BMI remains unclear because the MCI group in that study included more women than men. Another study found that APOE-ε4 is associated with weight loss in patients with AD, particularly women, but not in cognitively normal individuals, although the study included fewer than 50 patients with AD [6]. In addition, another long-term follow-up study of women showed that APOE-ε4 influences BMI change in older women, regardless of whether they have incident dementia or not [7]. Thus, the results of those studies uniformly indicate that APOE-ε4 is associated with weight loss in older women. However, studies in men are lacking. Further studies involving large datasets with both sexes based on robust statistical evidence are therefore required to elucidate the association between APOE-ε4 and body mass in patients with cognitive impairment. 

Another line of study indicated that specific components of body mass, i.e., fat-free mass (FFM), were shown to decrease with the progression of AD in patients with early-stage [8] and mild-to-moderate-stage AD [9], whereas there was increase in FM with the progression of AD.

Further evidence is therefore required to determine how APOE-ε4 contributes to decreases in FFM and/or FM at each stage of cognitive decline.

This study aimed to evaluate the association between APOE-ε4 and lower body mass, FFM, and FM in older adults with MCI and early-to-moderate stage AD using a large outpatient dataset.

## 2. Materials and Methods

### 2.1. Design and Study Population

The study subjects comprised outpatients (aged 65–89 years) who first presented to the Memory Clinic, National Center for Geriatrics and Gerontology (NCGG) located in Ohbu-shi, Japan, between October 2010 and February 2018, and were diagnosed with MCI and possible or probable AD based on the criteria of the National Institute on Aging-Alzheimer’s Association Workgroups [10,11]. Those with Barthel Index (BI) [12] scores < 80 and/or Mini-Mental State Examination (MMSE) [13] scores < 11 were excluded from the analysis. The stage of AD was categorized based on the MMSE scores as follows: MMSE ≥ 21, early; MMSE ≤ 20, moderate. The study protocol was approved by the ethics committee of the NCGG. Written informed consent was obtained from all participants before participation in the study.

### 2.2. Clinical Assessments

The body components FFM, FM, and percentage of body fat (% FM) were measured using bioelectrical impedance analysis (MC-180, Tanita Corp., Tokyo, Japan). 

Using the APOE-ε4 genotypes and phenotypes provided by the NCGG Biobank, the patients were divided into APOE-ε4 allele carriers (APOE4+) and non-carriers (APOE4−).

Demographic data, including age, years of education, smoking status, drinking status, dementia behavior disturbance scale (DBD) scores [14,15], and comorbidities (hypertension, dyslipidemia, and diabetes mellitus) were obtained from participants’ caregivers using questionnaires. Biochemical variables were evaluated as part of the medical examination. Depressive mood was assessed using the 15-item geriatric depression scale (GDS-15) [16].

### 2.3. Statistical Analyses

The distribution of variables was examined, and all analyses were sex-stratified. Differences in continuous and categorical variables between the APOE-ε4 carriers and non-carriers were compared using the t-test and chi-square test, respectively. The interaction between APOE-ε4 carriers’ status and sex was assessed using analysis of covariance. To evaluate the association of APOE-ε4 on BMI, weight, and body composition, multiple regression analysis was performed, with age, education, GDS-15, DBD, drinking status, smoking status, hypertension, dyslipidemia, diabetes mellitus, height, and MMSE as potential covariates. The slope corresponding to each continuous variable was obtained as the beta coefficient. Categorical variables were entered as a set of binary variables, and their slopes were obtained as the beta coefficient. All analyses were performed using R studio version 4.1.4 (R Foundation for Statistical Computing, Vienna, Austria). Statistical significance was set at *p* < 0.05.

## 3. Results

### 3.1. Participant Characteristics

The present analysis included 1631 patients. Of these, 604 had MCI (364 women) and 1027 had mild-to-moderate AD (703 women). The APOE4+ participants included 211 participants with MCI (34.9% of the total number of participants with MCI) and 439 participants with AD (42.7% of the total number of participants with AD). There was no significant difference in the frequency of APOE4+ between the sexes (*p* > 0.6). The clinical profiles of the subjects, subdivided by sex and APOE4 status, are shown in Table 1.

For both sexes, the mean age and MMSE scores were significantly lower among the APOE4+ subjects than among the APOE4− subjects, whereas T-Chol and LDL-C were higher among the APOE4+ subjects than among the APOE4− subjects. 

Among women, there were significant differences in BMI, FFM, FM, and %FM between the APOE4+ and APOE4− carriers (all *p* < 0.05). None of the variables of body composition differed according to the APOE4 status among men.

### 3.2. Association of APOE-ε4 with Body Composition

After adjusting for all potential covariates (Table 2), APOE4+ was significantly associated with a lower BMI (β = −0.496 ± 0.215 kg/m^2^ per presence, *p* = 0.021), weight (β = −1.116 ± 0.468 kg per presence, *p* = 0.017), FM (β = −1.196 ± 0.401 kg per presence, *p* = 0.003), and %FM (β = −1.700 ± 0.539% per presence, *p* = 0.002) in women but not in men.

We conducted subgroup analyses to evaluate the impact of APOE4+ on body composition between older adults with MCI and AD (Table 3). Subgroup analyses demonstrated that APOE4+ was significantly associated with lower weight (β = −2.265 ± 0.830 kg per presence, *p* = 0.007) and FM (β = −1.892 ± 0.697 kg per presence, *p* = 0.007) in women with MCI but not in those with early- and moderate-stage AD (all *p* > 0.05).

## 4. Discussion

This study revealed that the presence of the APOE-ε4 allele was associated with lower body mass in women with cognitive impairment. Notably, the association with lower body mass mostly reflected the association with lower FM but not with FFM.

These results support previously reported findings [6,7] and provide robust evidence that elucidates the findings reported by Bell et al. that remained inconclusive [5]. The contribution of the APOE-ε4 allele as expressed by the beta coefficient was shown to be only −1.1 for both weight and FM, which is approximately equivalent to 10 years’ worth of changes in weight and FM in individuals, given the beta coefficients between age and weight or FM in this study (−0.158 kg/year, weight; −0.088 kg/year, FM: Table 2). 

According to the results of the subgroup analyses (Table 3), the associations between the presence of the APOE-ε4 allele and lower body mass and FM were clearer in women with MCI than in those with AD. This effect might partially be explained by the fact that the trend of FM decline inverts with the progress of AD [9]. Additionally, although no associations were found between the APOE4 status and body composition in men, a negative association was observed between APOE4 and FM, albeit not significantly, with the progression of AD. There might be a sex difference in the timing of APOE-ε4’s effect on body composition changes. 

Results explaining the sex differences in the relationship between the APOE-ε4 allele and measures of body composition could not been obtained in this study, and the reason for their presence remains unclear. There might be an interaction effect of the APOE-ε4 allele with sex hormones such as estrogen, although our study could not assess this. An interaction effect between estrogen and APOE-ε4 has been reported for cognitive decline and carotid atherosclerosis in older women [17]. In addition, the results of the subgroup analyses regarding the association between APOE-ε4 and measures of body composition in each stage of cognitive decline also imply that there may have been sex differences in the heterogeneity of the APOE gene polymorphisms, influencing the sex differences in the relationship between the APOE-ε4 allele and body composition. Future studies are needed to clarify the reason for the observed sex differences and the association between APOE-ε4 and measures of body composition in men.

Although our study did not explore the mechanisms underlying the association between APOE-ε4 and lower body mass or FM, some mechanisms have been identified. One possibility is that APOE-ε4 may contribute to changes in hypothalamic function, and consequently, to changes in eating behavior and/or whole-body energy metabolism due to hypothalamic neurodegeneration. A study with transgenic mice found that overexpression of the amyloid precursor protein was associated with hypothalamic leptin signaling dysfunction, increased energy expenditure, and, consequently, weight loss [18]. A further study indicated that insulin concentrations were lower in older APOE-ε4 carriers, including cognitively normal individuals and those with AD, than in APOE-ε4 non-carriers, despite comparable blood glucose concentrations between the groups [19], suggesting that the decreased body mass and FM in older adults with APOE-ε4 may be due to the lack of insulin anabolic reactions. Further studies examining the changes in whole-body energy metabolism, including energy expenditure and intake, are needed. 

Our study had several strengths and limitations. First, it was based on robust evidence, as it involved a large number of patients with MCI and AD, and analyses of sex-related differences were performed. Additionally, our study was the first to demonstrate the association between APOE-ε4 and body composition (i.e., FFM and FM) in cognitively impaired older adults. However, this study provided no longitudinal data or evidence of the mechanisms underlying the association between APOE-ε4 and lower FM in older women with cognitive impairment. In addition, as discussed above, the relatively smaller sample size for men than for women may have contributed to the lack of a more profound exploration of associations among men. Future studies are therefore needed to clarify the association between APOE-ε4 and body composition in men.

In conclusion, the present findings indicate that APOE-ε4 may contribute to the risk of a lower body mass, particularly FM, among women during the early stage of cognitive decline (i.e., MCI). The lower body mass associated with APOE-ε4 may represent a non-cognitive manifestation of AD. These results highlight the importance of ensuring sufficient energy and nutrient intake, as well as monitoring changes in body mass, in older women who are APOE-ε4 carriers to lower the risk of progressive cognitive decline and mortality. 

## Figures and Tables

**Table 1 nutrients-14-00539-t001:** Demographic and clinical characteristics of participants by APOE-ε4 carrier status.

	Women *n* = 1067			Men *n* = 564			
	APOE 4− *n* = 637 (59.7%)	APOE 4+ *n* = 430 (40.3%)	*p* Value	APOE 4− *n* = 344 (61.0%)	APOE 4+ *n* = 220 (39.0%)	*p* Value	*p* for Interaction by Sex
Diagnosis, *n* (%)			0.039			0.011	0.447
MCI, *n* = 604	236 (37.0)	128 (29.8)		157 (45.6)	83 (37.7)		
Early AD, *n* = 436	173 (27.2)	123 (28.6)		91 (26.5)	49 (22.3)		
Moderate AD, *n* = 591	228 (35.8)	179 (41.6)		96 (27.9)	88 (40.0)		
Age, year	79.1 ± 5.4	76.8 ± 5.6	<0.001	78.0± 5.8	76.5 ± 5.3	<0.001	0.225
Education, year	9.8 ± 2.0	10.3 ± 2.1	<0.001	11.2 ± 3.0	11.4 ± 2.7	0.252	0.224
MMSE, score	21.7 ± 4.3	20.8 ± 4.3	0.002	22.4 ± 4.4	21.1 ± 4.6	<.001	0.532
BI, score	97.9 ± 4.7	98.7 ± 3.7	<0.001	98.2 ± 4.0	98.5 ± 3.6	0.435	0.175
GDS-15, score	4.2 ± 3.0	3.8 ± 2.9	0.018	3.6 ± 2.9	3.3 ± 2.5	0.563	0.602
DBD, score	12.6 ± 9.0	13.7 ± 10.0	0.177	12.5 ± 9.0	13.1 ± 9.4	0.525	0.648
BMI, kg/m^2^	22.2 ± 3.5	21.7 ± 3.3	0.016	22.8 ± 3.1	22.5 ± 3.2	0.149	0.520
Height, cm	147.0 ± 5.7	148.3 ± 5.7	0.001	161.1 ± 6.6	161.8 ± 6.5	0.228	0.350
Weight, kg	48.1 ± 8.6	47.8 ± 7.8	0.846	59.2 ± 8.9	58.9 ± 9.0	0.635	0.903
FFM, kg	33.2 ± 3.5	33.9 ± 3.3	0.001	45.9 ± 5.7	46.1 ± 5.5	0.591	0.317
FM, kg	14.9 ± 6.6	13.9 ± 6.3	0.011	13.3 ± 5.5	12.8 ± 5.5	0.310	0.387
% Fat mass, %	29.8 ± 8.4	28.0 ± 8.6	0.001	21.8 ± 6.8	21.1 ± 6.8	0.239	0.198
Hypertension, *n* (%)	405 (63.6)	255 (59.3)	0.178	218 (63.4)	126 (57.3)	0.174	0.725
Dyslipidemia, *n* (%)	230 (36.1)	182 (42.3)	0.047	103 (29.9)	71 (32.3)	0.623	0.446
Diabetes mellitus, *n* (%)	124 (19.5)	77 (17.9)	0.576	83 (24.1)	54 (24.5)	0.990	0.647
Drinking status, *n* (%)			0.757			0.057	0.531
Never	495 (77.7)	339 (78.8)		146 (42.4)	94 (42.7)		
Ethanol <43.2 g/day	140 (22.0)	89 (20.7)		180 (52.4)	123 (55.9)		
Ethanol ≥43.2 g/day	2 (0.3)	2 (0.5)		18 (5.2)	3 (1.4)		
Current smoking, *n* (%)	14 (2.2)	12 (2.8)	0.679	45 (13.1)	23 (10.5)	0.423	0.186
Albumin, *n* = 1515	4.4 ± 0.3	4.4 ± 0.3	0.338	4.4 ± 0.4	4.4 ± 0.3	0.429	0.438
T-Chol, *n* = 1504	219.3 ± 36.3	227.4 ± 39.2	0.004	193.3 ± 35.0	204.6 ± 35.6	0.001	0.434
TG, *n* = 1509	127.7 ± 74.3	126.7 ± 79.0	0.291	129.0 ± 78.7	134.9 ± 78.9	0.172	0.418
HDL, *n* = 1508	63.2 ± 15.6	64.6 ± 15.3	0.113	55.4 ± 15.4	54.1 ± 14.6	0.466	0.121
LDL, *n* = 1499	124.9 ± 29.8	133.1 ± 33.5	0.001	110.1 ± 30.3	122.2 ± 31.8	0.001	0.255

Abbreviations: AD, Alzheimer’s disease; APOE, apolipoprotein E; BI, Barthel index; BMI, body mass index; DBD, dementia behavior disturbance scale; FFM, fat-free mass; FM, at mass; GDS-15, 15-item geriatric depression scale; HDL, high-density lipoprotein cholesterol; LDL, low-density lipoprotein cholesterol; MCI, mild cognitive impairment; MMSE, Mini-Mental State Examination; T-Chol, total cholesterol; TG, triglyceride.

**Table 2 nutrients-14-00539-t002:** Association of each potential predictive variable with body mass index and body composition.

	Women			Men		
	Beta	95% CI	*p* Value	Beta	95% CI	*p* Value
BMI						
APOE+	−0.496 ± 0.215	−0.917–−0.075	0.021	−0.244 ± 0.267	−0.768–0.281	0.362
Age	−0.055 ± 0.020	−0.095–−0.016	0.006	−0.021 ± 0.024	−0.068–0.026	0.374
Education	−0.112 ± 0.056	−0.221–−0.003	0.043	0.047 ± 0.046	−0.044–0.138	0.311
MMSE	0.069 ± 0.026	0.017–0.121	0.008	0.060 ± 0.031	−0.001–0.122	0.053
GDS-15	−0.020 ± 0.035	−0.088–0.048	0.563	0.013 ± 0.047	−0.080–0.107	0.777
DBD	0.005 ± 0.011	−0.017–0.028	0.646	−0.001 ± 0.015	−0.029–0.029	0.980
Drinking status						
≥43.2 g/d	−1.990 ± 1.663	−5.252–1.273	0.232	0.425 ± 0.687	−0.924–1.774	0.536
<43.2 g/d	0.329 ± 0.250	−0.161–0.818	0.188	0.347 ± 0.263	−0.169–0.863	0.187
never	ref			ref		
Smoking status	−0.732 ± 0.660	−2.027–0.563	0.268	−0.390 ± 0.393	−1.161–0.381	0.321
Hypertension	1.296 ± 0.216	0.871–1.720	<0.001	0.983 ± 0.263	0.465–1.500	<0.001
Dyslipidemia	0.374 ± 0.213	−0.043–0.791	0.079	1.025 ± 0.281	0.472–1.578	<0.001
Diabetes mellitus	1.460 ± 0.264	0.941–1.978	<0.001	0.713 ± 0.300	0.124–1.301	0.018
Weight						
APOE+	−1.116 ± 0.468	−2.034–−0.199	0.017	−0.602 ± 0.686	−1.950–0.746	0.381
Height	0.548 ± 0.044	0.463–0.633	<0.001	0.550 ± 0.054	0.444–0.656	<0.001
Age	−0.158 ± 0.047	−0.249–−0.066	0.001	−0.111 ± 0.064	−0.237–0.014	0.081
Education	−0.223 ± 0.121	−0.460–0.015	0.066	0.168 ± 0.120	−0.069–0.404	0.165
MMSE	0.163 ± 0.058	0.050–0.277	0.004	0.172 ± 0.080	0.014–0.330	0.033
GDS-15	−0.044 ± 0.076	−0.192–0.105	0.564	0.002 ± 0.122	−0.238–0.241	0.988
DBD	0.010 ± 0.025	−0.039–0.059	0.688	0.001 ± 0.038	−0.074–0.075	0.994
Drinking status						
≥43.2 g/d	−4.318 ± 3.618	−11.417–2.780	0.233	1.475 ± 1.769	−1.999–4.950	0.405
<43.2 g/d	0.805 ± 0.544	−0.262–1.871	0.139	1.165 ± 0.680	−0.171–2.502	0.087
never	ref			ref		
Smoking status	−1.599 ± 1.436	−4.417–1.220	0.266	−1.165 ± 1.011	−3.151–0.822	0.250
Hypertension	2.804 ± 0.471	1.879–3.728	<0.001	2.435 ± 0.679	1.102–3.786	<0.001
Dyslipidemia	0.860 ± 0.462	−0.048–1.767	0.063	2.447 ± 0.727	1.020–3.874	0.001
Diabetes mellitus	3.242 ± 0.576	2.112–4.373	<0.001	1.934 ± 0.772	0.417–3.450	0.013

FFM						
APOE+	0.078 ± 0.156	−0.228–0.385	0.616	−0.360 ± 0.391	−1.128–0.408	0.358
Height	0.388 ± 0.015	0.360–0.417	<0.001	0.418 ± 0.031	0.357–0.478	<0.001
Age	−0.070 ± 0.016	−0.101–−0.040	<0.001	−0.205 ± 0.036	−0.276–−0.133	<0.001
Education	−0.041 ± 0.040	−0.120–0.039	0.313	0.089 ± 0.069	−0.046–0.224	0.195
MMSE	0.050 ± 0.019	0.012–0.088	0.010	0.076 ± 0.046	−0.014–0.166	0.098
GDS-15	0.011 ± 0.025	−0.039–0.060	0.669	0.023 ± 0.070	−0.113–0.160	0.737
DBD	0.001 ± 0.008	−0.016–0.017	0.943	0.012 ± 0.022	−0.030–0.055	0.573
Drinking status						
≥43.2 g/d	−2.003 ± 1.209	−4.376–0.369	0.098	0.329 ± 1.008	−1.651–2.309	0.744
<43.2 g/d	0.187 ± 0.182	−0.170–0.543	0.304	0.596 ± 0.388	−0.166–1.357	0.125
never	ref			ref		
Smoking status	−0.610 ± 0.480	−1.552–−0.332	0.204	−1.063 ± 0.576	−2.195–0.069	0.066
Hypertension	0.726 ± 0.157	0.417–1.035	<0.001	0.960 ± 0.387	0.200–1.720	0.013
Dyslipidemia	0.189 ± 0.155	−0.114–0.492	0.222	1.276 ± 0.414	0.463–2.090	0.002
Diabetes mellitus	0.692 ± 0.193	0.314–1.070	<.001	0.192 ± 0.440	−0.672–1.056	0.662
FM						
APOE+	−1.196 ± 0.401	−1.984–−0.408	0.003	−0.247 ± 0.473	−1.171–0.676	0.602
Height	0.159 ± 0.037	0.086–0.233	<0.001	0.133 ± 0.037	0.059–0.206	<0.001
Age	−0.088 ± 0.040	−0.166–−0.009	0.030	0.094 ± 0.044	0.007–0.180	0.034
Education	−0.181 ± 0.104	−0.385–0.022	0.081	0.078 ± 0.083	−0.085–0.241	0.347
MMSE	0.114 ± 0.050	0.016–0.211	0.022	0.096 ± 0.055	−0.012–0.205	0.083
GDS-15	−0.055 ± 0.065	−0.182–0.073	0.401	−0.021 ± 0.084	−0.186–0.144	0.800
DBD	0.010 ± 0.022	−0.033–0.052	0.654	−0.011 ± 0.026	−0.063–0.039	0.647
Drinking status						
≥43.2 g/d	−2.338 ± 3.106	−8.433–3.757	0.452	1.146 ± 1.220	−1.999–4.950	0.348
<43.2 g/d	0.617 ± 0.467	−0.299–1.533	0.186	0.571 ± 0.469	−0.171–2.502	0.224
never	ref			ref		
Smoking status	−0.994 ± 1.233	−3.414–1.426	0.420	−0.093 ± 0.698	−1.462–1.276	0.894
Hypertension	2.073 ± 0.404	1.582–2.867	<.001	1.473 ± 0.468	0.557–2.388	0.002
Dyslipidemia	0.672 ± 0.397	−0.107–1.451	0.091	1.172 ± 0.501	0.189–2.156	0.020
Diabetes mellitus	2.553 ± 0.495	1.582–3.524	<0.001	1.744 ± 0.532	0.699–2.788	0.001
% Fat mass						
APOE+	−1.700 ± 0.539	−2.758–−0.643	0.002	−0.238 ± 0.592	−1.401–0.925	0.688
Age	−0.078 ± 0.054	−0.183–0.028	0.150	0.201 ± 0.055	0.092–0.309	0.000
Education	−0.259 ± 0.140	−0.532–0.015	0.064	0.042 ± 0.104	−0.163–0.246	0.689
MMSE	0.132 ± 0.067	0.001–0.263	0.048	0.110 ± 0.069	−0.026–0.246	0.114
GDS-15	−0.129 ± 0.087	−0.301–0.042	0.139	−0.021 ± 0.105	−0.228–0.186	0.841
DBD	−0.009 ± 0.029	−0.066–0.047	0.750	−0.016 ± 0.033	−0.080–0.048	0.628
Drinking status						
≥43.2 g/d	−1.994 ± 4.172	−10.180–6.193	0.633	1.508 ± 1.527	−1.491–4.507	0.324
<43.2 g/d	0.874 ± 0.629	−0.356–2.103	0.164	0.510 ± 0.587	−0.643–1.664	0.385
never	ref			ref		
Smoking status	−0.378 ± 1.656	−3.628–2.872	0.820	0.063 ± 0.873	−1.652–1.778	0.942
Hypertension	2.435 ± 0.543	1.369–3.501	<.001	1.543 ± 0.586	0.392–2.694	0.009
Dyslipidemia	0.920 ± 0.533	−0.127–1.966	0.085	1.178 ± 0.627	−0.054–2.409	0.061
Diabetes mellitus	2.789 ± 0.665	1.485–4.093	<0.001	2.020 ± 0.666	0.711–3.329	0.003

Abbreviations: APOE, apolipoprotein E; BI, Barthel index; BMI, body mass index; CI, confidence interval; DBD, dementia behavior disturbance scale; FFM, fat-free mass; FM, fat mass; GDS-15, 15-item geriatric depression scale; MMSE, mini-mental state examination.

**Table 3 nutrients-14-00539-t003:** Impacts of APOE-ε4 on measures of body composition in each stage of cognitive decline.

	MCI		Early AD		Moderate AD	
	Beta	*p* Value	Beta	*p* Value	Beta	*p* Value
Women						
Weight	−2.265 ± 0.830	0.007	−0.669 ± 0.916	0.465	−0.454 ± 0.736	0.538
FFM	−0.374 ± 0.271	0.168	0.280 ± 0.292	0.339	0.106 ± 0.258	0.682
FM	−1.892 ± 0.697	0.007	−0.946 ± 0.798	0.237	−0.564 ± 0.637	0.376
%Fat mass	−2.462 ± 0.914	0.007	−1.323 ± 1.060	0.213	−0.983 ± 0.877	0.263
Men						
Weight	0.800 ± 1.034	0.440	−0.603 ± 1.1495	0.687	−2.151 ± 1.292	0.098
FFM	−0.227 ± 0.608	0.709	−0.280 ± 0.292	0.339	−0.957 ± 0.737	0.196
FM	1.022 ± 0.692	0.141	−0.330 ± 1.067	0.757	−1.196 ± 0.885	0.179
%Fat mass	1.260 ± 0.863	0.649	−0.394 ± 1.336	0.769	−1.148 ± 1.106	0.301

These variables were adjusted for age, education, GDS-15, DBD, drinking status, smoking status, hypertension, dyslipidemia, diabetes mellitus, height, and MMSE. Abbreviation: AD, Alzheimer’s disease; BMI, body mass index; FFM, fat-free mass; FM, fat mass; MCI, mild cognitive impairment.

## Data Availability

Anonymized data will be available on request from any qualified investigator after clearance by the ethics committee.

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
