# Peer review of "ApoE4 Is Associated with Lower Body Mass, Particularly Fat Mass, in Older Women with Cognitive Impairment"

_nutrients, 2022, doi:10.3390/nu14030539_

Round 1
Reviewer 1 Report
Ando et al reported that APOE-ε4 carriers with mild cognitive impairment and early- to moderate-stage AD had lower weight, fat mass, body fat in women but not men. This paper is generally well written.
Perhaps in the discussion of the paper the authors could mention the interaction between estrogen and APOE-ε4. Since the lower weight, fat mass, and body fat were found only in women and not men, the authors should mention the role of estrogen in AD and APOE-ε4. Also, the authors may want to mention that in post menopause women, the incidence of AD increases.
Reviewer 2 Report
While this manuscript is interesting and provides strong data regarding the association of APOE-e4 from a large cohort, there is a gap in the analysis that should be addressed before publication. The authors failed to look at potential differences between APOE-e4 homozygous and heterozygous individuals. If this data is unavailable, the authors should at least discuss the potential impact of 1 vs 2 e4 alleles on BMI and fat mass with AD and mention that should be explored in future research. However, if the data is available, the authors should conduct this analysis and add it to the resubmitted manuscript. I believe this additional data will further strengthen interest in this research.
It is also suggested that in the methods section, more information should be provided on the beta coefficient and how these data are calculated. Also, in Tables 2 and 3, the column heading have "B". This should be "beta" for clarity to the reader.
Round 2
Reviewer 2 Report
I don't feel that the authors addressed my major issue with my review in their revised manuscript. They did not show data on APOE e4 homozygotes vs heterozygotes. If this data isn't available, the authors should explain why. I don't feel that this paper should be published in it's current form.